# Comparison of perinatal outcome and mode of birth of twin and singleton pregnancies in migrant and refugee populations on the Thai Myanmar border: A population cohort

Taco J. Prins[1,2,3]*, Aung Myat Min[4], Mary E. Gilder[4], Nay Win Tun[4], Janneke Schepens[4], Kathryn McGregor[4], Verena I. Carrara[4,5], Jacher Wiladphaingern[4], Mu Koh Paw[4†], Eh Moo[4], Julie A. Simpson[6], Chaisiri Angkurawaranon[1,2], Marcus J. Rijken[4,7], Michele van Vugt[4,3], François Nosten[4,8], Rose McGready[4,8]

1 Department of Family Medicine, Faculty of Medicine, Chiang Mai University, Chiang Mai, Thailand, 2 Global Health and Chronic Conditions Research Group, Chiang Mai University, Chiang Mai, Thailand, 3 Amsterdam University Medical Centres, Department of Internal Medicine & Infectious diseases, and Research groups: APH, GH and AII&I, Amsterdam UMC, Amsterdam, The Netherlands, 4 Shoklo Malaria Research Unit, Mahidol–Oxford Tropical Medicine Research Unit, Faculty of Tropical Medicine, Mahidol University, Mae Sot, Thailand, 5 Institute of Global Health, Faculty of Medicine, University of Geneva, Geneva, Switzerland, 6 Centre for Epidemiology and Biostatistics, Melbourne School of Population and Global Health, The University of Melbourne, Melbourne, Australia, 7 Department of Obstetrics and Gynaecology, Amsterdam UMC, Amsterdam, The Netherlands, 8 Centre for Tropical Medicine and Global Health, Nuffield Department of Medicine, University of Oxford, Oxford, United Kingdom

† Deceased.
* taco.prins@cmu.ac.th

**Data Availability Statement:** Data cannot be shared publicly because this is a population of undocumented refugees and migrants and we do

## Abstract

### Background

In low- and middle-income countries twin births have a high risk of complications partly due to barriers to accessing hospital care. This study compares pregnancy outcomes, maternal and neonatal morbidity and mortality of twin to singleton pregnancy in refugee and migrant clinics on the Thai Myanmar border.

### Methods

A retrospective review of medical records of all singleton and twin pregnancies delivered or followed at antenatal clinics of the Shoklo Malaria Research Unit from 1986 to 2020, with a known outcome and estimated gestational age. Logistic regression was done to compare the odds of maternal and neonatal outcomes between twin and singleton pregnancies.

### Results

Between 1986 and 2020 this unstable and migratory population had a recorded outcome of pregnancy of 28 weeks or more for 597 twin births and 59,005 singleton births. Twinning rate was low and stable (<9 per 1,000) over 30 years. Three-quarters (446/597) of the twin pregnancies and 96% (56,626/59,005) of singletons birthed vaginally. During pregnancy, a significantly higher proportion of twin pregnancies compared to singleton had pre-eclampsia

not have their permission to share their data. Data are available from the Mahidol-Oxford Research Unit Institutional data access committee upon reasonable request from researchers who meet the criteria for access to confidential data (contact Rita Chanviriyavuth, email rita@tropmedres.ac).

**Funding:** This project is partially supported by Chiang Mai University to TJP and CA. JAS is supported by an Australian National Health and Medical Research Council Leadership Investigator Grant (#1196068). The others authors received no funding for this study. The Shoklo Malaria Research Unit is supported in part by the Wellcome-Trust Major Overseas Programme in Southeast Asia (# 220211, https://doi.org/10.35802/220211; lead applicant Nicholas Day). For the purpose of Open Access, the author has applied a CC BY public copyright licence to any Author Accepted Manuscript version arising from this submission. There was no additional external funding received for this study. The funders had no role in study design, data collection and analysis, decision to publish, or preparation of the manuscript.

**Competing interests:** The authors have declared that no competing interests exist.

**Abbreviations:** LMICs, Low and Middle income countries; OR, Odds Ratio; PPH, postpartum hemorrhage; ANC, antenatal care; SMRU, Shoklo Malaria Research Unit; TBA, traditional birth attendant; EGA, estimated gestational age; APH, antepartum haemorrhage; SGA, Small for gestational age; CI, confidence interval; EmOC, Emergency obstetric Course; ALSO, Advanced Life Support in Obstetrics.

(7.0% versus 1.7%), gestational hypertension (9.9% versus 3.9%) and eclampsia (1.0% versus 0.2%). The stillbirth rate of twin 1 and twin 2 was higher compared to singletons: twin 1 25 per 1,000 (15/595), twin 2 64 per 1,000 (38/595) and singletons 12 per 1,000 (680/58,781). The estimated odds ratio (95% confidence interval (CI)) for stillbirth of twin 1 and twin 2 compared to singletons was 2.2 (95% CI 1.3–3.6) and 5.8 (95% CI 4.1–8.1); and maternal death 2.0 (0.95–11.4), respectively, As expected most perinatal deaths were 28 to <32 week gestation.

## Conclusion

In this fragile setting where access to hospital care is difficult, three in four twins birthed vaginally. Twin pregnancies have a higher maternal morbidity and perinatal mortality, especially the second twin, compared to singleton pregnancies.

## Introduction

The worldwide twinning rate has increased from 9.1 per 1,000 births in 1980–1985 to 12.0 in 2010–2015 [1]. Contributions to this rise include older maternal age and assisted reproduction, the latter of which is not available in marginalized settings [1, 2].

Compared to singleton pregnancies, twin pregnancies have a higher risk of poor outcomes, such as postpartum hemorrhage (PPH) and preterm birth, as well as obstetrical and post-partum complications [3–5]. Generally the second twin has a higher risk of morbidity and mortality because of obstetric complications that may occur after birth of the first twin, including placental separation, cord prolapse, uterine atony [6–8]. Different approaches are proposed for the mode of birth for twin pregnancies and studies report contradictory results on the risk of perinatal and neonatal mortality for the second twin when birthed vaginally or through a planned caesarean delivery at term or preterm [9–13]. The general obstetric consensus is that vaginal birth is safe when the first twin is in cephalic position [2, 12, 14, 15].

In low- and middle-income countries (LMICs) access to care, during pregnancy and at birth, is a significant barrier for women carrying multiple or singleton births, compared to high income countries. Access to care during this part of the life course are elevated in multiples compared to singletons, due to the higher risk of complications: mortality and morbidity [4, 5]. As the background health of women in LMIC (e.g. anaemic at the time of labor) and as maternal and neonatal mortality following caesarean delivery is disproportionally high in LMICs, the woman's choice of the mode of birth is important [16–18]. For women in migrant populations the choice is frequently limited due to barriers relating to language, culture, cost, documentation, and lack of health insurance [19, 20]. A caesarean delivery has a higher risk for complications if there are subsequent pregnancies and births; with the risk exacerbated in countries with unstable health care systems due to conflict [21]. Communities in the Thai-Myanmar border area have been conflict-affected for decades, which has significantly impacted health care service provision and outcomes. Information about management and outcomes of multiple pregnancy in marginalized populations such as this is a significant gap in the evidence-base for recommendations in multiple pregnancy [22].

The objective of this study was to assess maternal and neonatal complications and pregnancy outcomes of twin pregnancy and mode of birth in a refugee and migrant setting on the

Thai Myanmar border and compare their outcomes to singleton pregnancies within the same period.

## Methods

### Ethical approval

Ethical approval for retrospective analysis of hospital records at the Shoklo Malaria Research Unit was provided by the Oxford University Ethics Committee (OXTREC: 28–09), by Chiang Mai University (FAM-2566-0162) and the local community advisory board in Mae-Sot, Thailand (TCAB-4/1/2015).

### Study design

A retrospective review of medical records of all singleton and twin pregnancies delivered or who followed antenatal care (ANC) at Shoklo Malaria Research Unit (SMRU), during the period from 1986 to 2020 was performed. The STROBE guideline for observational studies was followed for reporting [23]. The clinical details were extracted from medical records which were initially paper based and became electronic in late 2008. Paper-based records prior to 2008 were extracted manually to an electronic database so that ANC and birth outcome data could be retrieved electronically. Information from records such as the paper-based partogram or the hospital referral forms was retrieved from archives stored at SMRU when required for clarification. Data extraction was completed in August 2022. All identifiable information of the women was removed or anonymised.

### Setting and background

On the Thai-Myanmar border, Tak Province, the SMRU has established a system of antenatal clinics initially in refugee camps commencing in 1986 (ceasing in 2016) and for migrants commencing in 1998. Marginalized populations of Burmese and Karen, and less frequently other ethnic groups, are served by the SMRU clinics. Initially these clinics were established to provide active detection and early treatment of malaria, which was the leading cause of death in the 1980s [24]. The impact of geography on access to ANC and pregnancy outcomes, including twin pregnancies, in the rural, remote and politically complex Thai-Myanmar border region has been detailed previously [25]. From 1986–2000, gestational age was estimated from fundal height at first ANC or neonatal exam (Dubowitz) after birth. Since late 2001, ultrasound for gestational age estimation became available in the clinics [26]. The SMRU trains local midwifes with its own curriculum of 15 months and provides short emergency obstetric care (EmOC) courses such as Advanced Life Support in Obstetrics (ALSO) [27]. The ALSO course includes training in labour dystocia, shoulder dystocia, (pre) eclampsia, breech and malpresentation, umbilical cord prolapse, preterm labour, ante and postpartum haemorrhage, first-trimester bleeding, maternal and neonatal resuscitation, severe malaria and sepsis [27].

Women are encouraged to deliver with trained midwives in the SMRU delivery units, instead of giving birth at home with a traditional birth attendant (TBA). WHO partographs to monitor labour were introduced in 1994 and included recording of fetal distress, meconium-stained amniotic fluid, tachysystole (excessive uterine activity), uterine rupture, abnormal maternal blood pressure during labour, mode of birth, Apgar score, neonatal resuscitation and postpartum haemorrhage (PPH) [27, 28]. Complicated births requiring caesarean delivery are referred to a local Thai hospital (15 minute to one hour drive, depending on the location of the clinic) with a 24 hour standby car ready to support these cases. When a multiple gestation was diagnosed, ultrasound scans were offered more frequently, every 2 weeks from 28 weeks for

fetal growth and at 34 and 36 weeks for position and sizes of the fetus, if the woman was able to access the clinic. The estimated date of delivery of twins was calculated from the earliest scan using the largest measured fetus. The general rule and advice for childbirth with twin pregnancy has been to attempt vaginal birth if there were no complications, the first fetus was cephalic or it was a premature labour. The birth units are midwife led with twin births co-supervised by an obstetric doctor, trained in multiple deliveries. The doctor is not present for the full 24 hours but may stay to assist if twin delivery is expected. Communication through radio was replaced by phone contact in 1996. In general after birth of the first twin, the second twin would be brought into a longitudinal position through external and rarely internal version if necessary. After twenty minutes or in cases requiring expedited delivery e.g. fetal distress, artificial rupture of the membranes and/or oxytocin would be commenced. Women who did not meet these criteria were referred to the nearest Thai Public Hospital for booking for caesarean delivery where the language spoken is Thai, which differs from the Karen and Burmese languages of the patient population.

From 1992 to 2008 dexamethasone was given at 28 weeks as a prophylactic dose to all twins, 8mg intra-muscularly eight hourly for 24 hours and when preterm labour was diagnosed. This was discontinued in 2009 and the drug was given only in the event of suspected or confirmed premature labour. During pregnancy all women with multiple gestation were treated for anaemia (except when haematocrit is higher than 40%), this included daily ferrous sulphate, folic acid, vitamin C and vitamin B12. During singleton pregnancy, women were treated for anaemia if the haematocrit was below 30%, otherwise women received anaemia prophylaxis with low dose ferrous sulphate and folic acid. In this setting, only oxygen, intravenous fluid, antibiotics, nasogastric feeding and phototherapy have been available in the special care baby unit (SCBU) from 2008 for the support of unwell preterm infants and reducing neonatal mortality by half [29]. SCBU improved the level of care available prior to 2008: oxygen, intramuscular antibiotics and nasogastric feeding. A neonatal intensive care unit (NICU) was not available routinely for this population. Midwifes of the SMRU were trained in Apgar and routinely assessed newborns from 1998 and this data was not provided on discharge from births in Thai Hospitals.

### Participants

All women with singleton or twin pregnancies who delivered and/or attended SMRU ANCs between 1986 and 2020 were included. Pregnancies were excluded if the outcome (a high loss to follow-up remains a constant feature of this mobile, conflict affected and marginalized population) or EGA was unknown, or if EGA at birth was <28 weeks. The reason extreme prematurity (EGA <28 weeks) was excluded was because in this, as in other limited resource settings, there is only active management from an EGA of 28 weeks. A previous publication from the same population reviewing 21 years of birth outcomes from 22 to ≤28 weeks identified extreme prematurity as representing less than 1% of birth outcomes, a 1 year survival of less than 1% and an over-representation of twins [30]. Rare events including conjoined twin and papyraceus twin pregnancies were excluded [31]. Women identified with a triplet pregnancies were excluded as the number was small (n = 9). Missing data per variable extracted is presented in the study flow (Fig 1).

### Variables

Anaemia was defined as haematocrit less than 30% measured during pregnancy. Maternal blood pressure was defined as abnormal if it was equal or above 140/90 mmHg. Preterm labour was defined as EGA <37 weeks. EGA was determined by ultrasound (2001 to 2020,

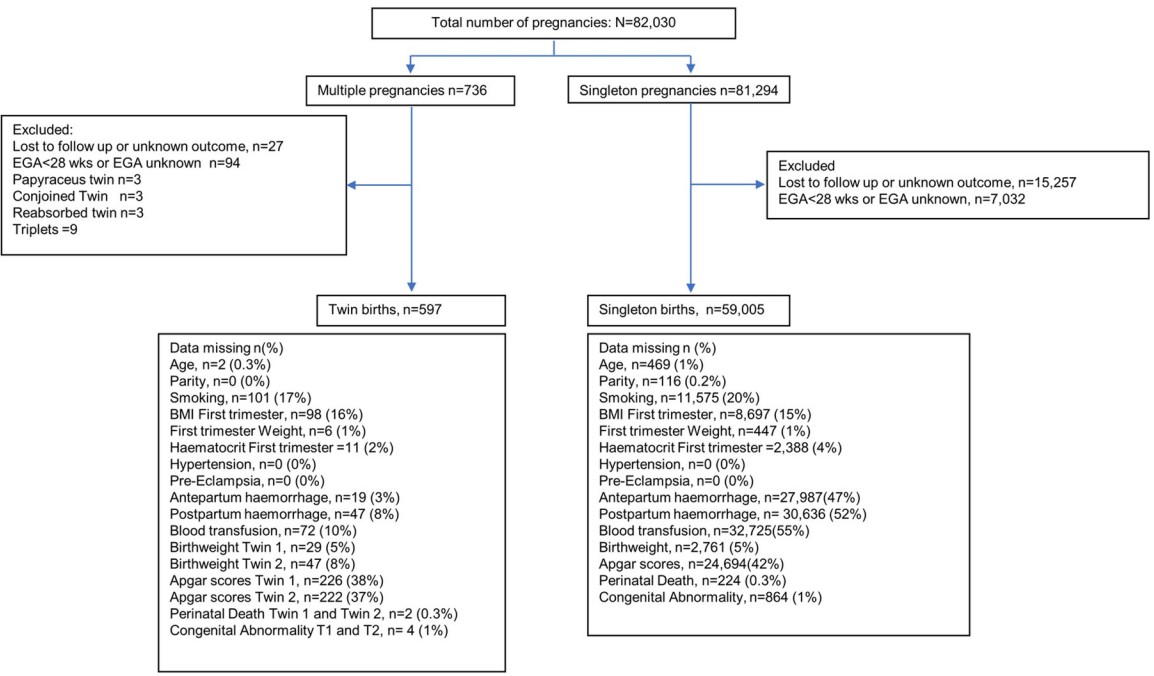

**Fig 1. Flow chart of data available and missing values.**

chorionicity was not available). When ultrasound was not available (prior to 2001), Dubowitz Assessment of gestation after birth, a population specific model for women with multiple symphysis fundal height measurements, last menstrual period or by the woman's reported self-estimate at first ANC. Antepartum haemorrhage (APH) was defined as antepartum blood loss (EGA ≥28 weeks) and postpartum haemorrhage was defined as blood loss ≥500 ml after birth. Small for gestational age (SGA) was defined as <10th percentile of birthweight for gestational age using the international standards of the INTERBIO-21st Project [32]. Congenital abnormalities were defined as any abnormality observed based on systematic physical exam, routinely since 1998. BMI and gestational weight gain were determined in all women at first ANC, but analysis was limited to women who attended in the first trimester, to have a baseline weight with minimal influence of the weight of the twin or singleton gestation. Twinning rate was defined as twin birth per 1,000 total births and was determined for 5-year intervals between 1986–2020. The use of dexamethasone for maturation of fetal lungs was provided as treatment in cases of threatened preterm labor <34 weeks up to 2013 and <35 weeks from 2014, and occasionally as prophylaxis based on the clinical presentation e.g. twin pregnancy in a woman with a history of preterm labor. Preterm birth was divided in two categories, 28 till <32 weeks EGA (very preterm) and 32 weeks till <37 weeks EGA (moderate and late preterm). Stillbirth rate was defined as number of stillbirths per 1,000 total births from EGA ≥28 weeks. Early Neonatal Death rate was defined as neonatal death in the first 7 days after birth per 1,000 live births from EGA≥28 weeks with the assumption that if the death was not recorded the infant was still alive [33]. Stillbirth and Early Neonatal Death rate were graphed from 28 to 42 weeks EGA at birth for the 35-year period 1986–2020. Stillbirth and Early Neonatal Death rate were described over three time periods 1990–2000, 2001–2010 and 2011–2020 to observe for change. Maternal death was defined as death of the mother during pregnancy and up to 6 weeks postpartum, with the caveat that it was with the same inclusion as the original cohort i.e. women who gave birth from 28 weeks with a known EGA.

## Statistical analysis

Data were analyzed using R (version 4.2.1) for Windows. Continuous data were described using the mean, standard deviation minimum, and maximum. Categorical data were summarized using frequency and proportions. Multivariable logistic regression was performed to compare maternal complication outcomes between twin and singleton pregnancies, adjusting for the potential confounders: maternal age, parity and smoking. For perinatal and neonatal outcomes, the frequency and proportion were presented for twin 1, twin 2 and singletons. Unadjusted and adjusted Odds Ratios (ORs; 95% Confidence Intervals (CI)) were also presented for these outcomes comparing twin 1 and twin 2 to singletons, with adjustment for the confounders: maternal age, parity, gestational hypertension, pre-eclampsia, eclampsia and smoking. There is no adjustment for SGA or preterm birth as this is an outcome of twin pregnancy.

## Results

Between 1986 and 2020, there were 81,294 singletons and 736 (1%, 736/81,294) multiple pregnancies, registered to SMRU ANC (Fig 1). The twinning rate was relatively stable over time, 5 (95%CI 3–10) per 1,000 births in the period from 1986–1990, and 9 (95%CI 7–11) per 1,000 births from 2016–2020 (S1 Table in the S1 File). In this unstable and migratory population 59,602 had a recorded outcome of birth of with an estimated gestation of 28 weeks or more (twin 597, singleton 59,005).

## Maternal characteristics

Compared to women with singleton pregnancies, women with twin pregnancies were older, multiparous and more likely to be smokers (Table 1). For women who attended their first ANC in first trimester, the mean weight and BMI were similar in twin and singleton pregnancies. The mean gestational weight gain was higher in twin pregnancy, 9.3kg (standard deviation SD = 5.6) compared with singleton pregnancies, 8.0kg (SD = 4.5). During pregnancy, 51.2% (287/561) of the mothers with a twin pregnancy were found to be anaemic compared to 37.4% (20,694/55,265) of the singletons.

**Table 1. Distribution of maternal characteristics for twin and singleton pregnancies.**

| Maternal characteristics | Twin pregnancies (n = 597) | Singleton pregnancies (n = 59,005) |
|---|---|---|
| Age (years) at first antenatal visit, Mean (SD) [min-max] | 28.5 (6.5) [15–45] | 26.2 (6.6) [13–53] |
| Nulliparous | 20.3 (121/597) | 31.7 (18,664/58,889) |
| Smoker | 33.5 (166/496) | 21.6 (10,224/47,430) |
| ANC in First trimester | 43.4 (259/597) | 43.5 (25,648/59,005) |
| First trimester; mean (SD) [min-max] Weight (kg) BMI (kg/m$^2$) | 48.4 (8.0) [31–76] 21.3 (3.3) [15.4–32.2] | 48.0 (7.7) [28–108] 21.2 (3.2) [13.5–45.1] |
| Weight gain (kg) first trimester to birth; mean (SD) [min-max] | 9.3 (5.6) [-7 to 24] | 8.0 (4.5) [-7.4 to 29] |
| Haematocrit (%) at first antenatal visit first trimester; Mean (SD) [min-max] | 34.5 (4.1) [22–47] | 35.6(3.8) [12–52] |
| Anaemic during pregnancy Nulliparous Multiparous | 51.2 (287/561) 35.0 (41/117) 55.4 (246/444) | 37.4 (20,694/55,265) 30.0 (5,277/17,579) 40.9 (15,391/37,599) |

Data are % (n/N) unless otherwise stated; SD—standard deviation

**Table 2. Obstetric complications in twin and singleton pregnancies and adjusted odds in twin compared to singletons.**

| Maternal complication outcome % (n) | Twin | Singleton | adjOR (95% CI)[a] |
|---|---|---|---|
| Estimated Gestational Age (weeks) at birth (SD); mean [min-max] | 36.5 (2.9) [28–43] | 38.9 (1.9) [28–43] | NA |
| Preterm labour<br>  28+0 to <32 weeks<br>  32+0 to <37 weeks<br>Term labour ≥37 weeks | 7.9 (47/597)<br>36.0 (215/597)<br>56.1 (335/597) | 1.1 (623/59,005)<br>8.3 (4,897/59,005)<br>90.6 (53,485/59,005) | 12.5 (8.6–18.1)<br>10.0 (8.2–12.1)<br>Reference outcome |
| Gestational hypertension | 9.9 (59/597) | 3.9 (2,289/59,005) | 2.4 (1.8–3.2) |
| Pre-eclampsia | 7.0 (42/597) | 1.7 (997/59,005) | 3.9 (2.7–5.3) |
| Eclampsia | 1.0 (6/597) | 0.2 (136/59,005) | 6.0 (2.3–12.7) |
| Antepartum haemorrhage | 1.4 (8/578) | 1.2 (363/31,018) | 1.3 (0.6–2.4) |
| Postpartum haemorrhage | 19.3 (106/550) | 6.7 (1,894/28,369) | 3.4 (2.7–4.3) |
| Transfusion for obstetric haemorrhage | 7.2 (38/525) | 1.7 (437/26,280) | 5.1 (3.5–7.2) |

Data are % (n/N) unless otherwise stated.

[a] Adjusted for maternal age, parity and smoking

**Morbidity during pregnancy and labour.** There was a higher proportion of obstetric complications during twin compared to singleton pregnancies (Table 2). After adjustment for maternal age, parity and smoking the adjusted odds (adjOR) of preterm labour for twins compared to singletons was 12.5 (95%CI: 8.6–18.1) for EGA from 28 to <32 weeks, and 10.0 (95% CI: 8.2–12.1) from 32 to <37 weeks. As expected, hypertensive disorders of pregnancy, and PPH were all more prevalent for women with twin compared to singleton pregnancies (Table 2).

**Singleton and twin births.** Mean gestational age at birth for twin pregnancies was 36.5 weeks (SD = 2.9), compared to 38.9 weeks (SD = 1.9) for singleton (Table 2). Three quarters, 75% (446/597), of the women with a twin pregnancy, birthed both babies vaginally, compared to 96% (56,626/59,005) for singletons (Table 3). Of women in the caesarean delivery group with a known position of twin 1, 61% (27/44) were non-cephalic. The percentage of caesarean delivery deliveries increased over time for twin and singleton pregnancies. For twin 1 and twin 2 the proportions in 1991–2000 were 13% (18/137) and 14% (19/137) and in 2011–2020 29% (63/219) and 34% (74/219) respectively; while caesarean deliveries for singletons in the same time periods were 2% (291/13,773), and 6% (1,331/21,960).

## Perinatal and neonatal outcome

The mean birthweight was 2,214grams (SD = 528) for the first twin delivered, 2,124 grams (SD = 528) for twin 2 and 2,946 grams (SD = 482) for singletons (Table 4). A high proportion of newborns were SGA: 51.9% (295/568) for twin 1, 60.9% (335/550) for twin 2 and 22.6%

**Table 3. Mode of birth for twin1, twin 2 and singletons.**

| Mode of birth | Twin pregnancy | | Singleton pregnancy (n = 59,009) |
|---|---|---|---|
| | Twin one | Twin two | |
| Birthed Vaginally | 78.2 (467/597) | 74.7 (446/597) | 96.0 (56,626/59,005) |
| Birthed by caesarean delivery | 21.4 (128/597) | 25.0 (149/597) | 4.0 (2,365/59,005) |

Data are % (n/N)

**Table 4. Perinatal and neonatal outcomes for twin 1 and twin 2 and singletons.**

| | | | Twin pregnancy | | Singleton pregnancy | OR (95% CI) compared to singletons | |
| --- | --- | --- | --- | --- | --- | --- | --- |
| | | | Twin 1 | Twin 2 | | Twin 1 | Twin 2 |
| Birthweight (g), Mean (SD) [Min–Max] | | | 2,214 (528) [710–4,000] | 2,124 (528) [700–3,900] | 2,946 (482) [520–5,580] | NA | NA |
| Congenital abnormality % (n/N) | | | 1.2 (7/593) | 1.4 (8/593) | 1.7 (970/58,141) | Unadjusted 0.7 (0.3–1.4) Adjusted 0.7 (0.3–1.3)[a] | Unadjusted 0.8 (0.4–1.5) Adjusted 0.8 (0.3–1.4)[a] |
| Small for gestational age % (n/N)) | | | 51.9 (295/568) | 60.9 (335/550) | 22.6 (12,699/56,244) | Unadjusted 3.7 (3.1–4.4) Adjusted 3.6 (3.0–4.4)[a] | Unadjusted 5.3 (4.5–6.4) Adjusted 4.9 (4.0–5.9)[a] |
| APGAR <7 at 5 minutes[b] % (n/N) | | | 1.7 (6/354) | 2.1 (7/335) | 1.5 (485/33,407) | Unadjusted 1.0 (0.4–2.3) Adjusted 1.2 (0.5–2.4)[a] | Unadjusted 1.4 (0.6–2.8) Adjusted 1.4 (0.6–2.8)[a] |
| Perinatal Mortality | Stillbirth rate[c] | N | 25 per 1,000 (15/595) | 64 per 1,000 (38/595) | 12 per 1,000 (680/58,781) | Unadjusted 2.2 (1.3–3.6) Adjusted 2.3 (1.3–3.8)[a] | Unadjusted 5.8 (4.1–8.1) Adjusted 5.7 (3.8–8.1)[a] |
| | Early Neonatal Death rate[b,d] | N | 48 per 1,000 (28/580) | 50 per 1,000 (28/557) | 10 per 1,000 (574/58,101) | Unadjusted 5.1 (3.4–7.4) Adjusted 4.1 (2.4–6.5)[a] | Unadjusted 5.3 (3.5–7.7) Adjusted 4.3 (2.5–6.7)[a] |
| Maternal Death % (n/N) | | | 0.50 (3/597) | | 0.11 (65/59,005) | Unadjusted 4.6 (1.1–12.4) Adjusted 4.0 (0.95–11.4)[a] | |

[a] Adjusted for maternal age, parity, gestational hypertension, pre-eclampsia, eclampsia, smoking

[b] Stillbirth excluded

[c] Stillbirth per 1,000 total births

[d] Early Neonatal Death per 1,000 live births

(12,699/56,244) for singletons. As expected the risks of SGA in twins were high compared to singletons; adjOR 3.6 (95%CI: 3.0–4.4) for twin 1 and adjOR 4.9 (95% CI: 4.0–5.9) for twin 2.

## Stillbirth

Compared to singletons, twin 1 (adjOR 2.3, 95% CI: 1.3–3.8) and twin 2 (adjOR 5.7, 95% CI 3.8–8.1) were at greater risk of stillbirth (Table 4). The stillbirth nadir occurred in the 37–38 week gestational age window, after which the stillbirth rate for twins increased, in comparison with the singletons where it remains stable from 37–40 weeks (Fig 2). Most of the twin 2 stillbirths occurred during vaginal birth (28 out of 36) with the remainder occurring during caesarean delivery, from which 6 were known emergency caesarean delivery. The position of twin one, when twin 2 had a stillbirth, was known in 30 cases, 26 were in vertex position and 4 in non-vertex. Any discordance greater than 40% between twins resulted in an excess of stillbirth of the smaller twin (S2 Table in the S1 File). The stillbirth rate was highest outside the hospital and SMRU (S3 Table in the S1 File).

## Early Neonatal Death

There was a four-fold increased odds in early neonatal death for twin births compared to singletons, adjOR 4.1 (2.4–6.5) for twin 1 and 4.3 (2.5–6.7) for twin 2 (Table 4). The early neonatal death rate was highest in premature neonates in both twins and singletons and was independent of mode of birth of twin 1 (Fig 3, S4 and S5 Tables in the S1 File). The early neonatal mortality decreased over time for both twins and singletons (Table 5).

Of first and second twins with early neonatal death: 75% (21/28) occurred at EGA<35 weeks at birth and 3 of their mothers received prophylactic doses of dexamethasone the week before birth. For twin 2, 2 out of 21 early neonatal deaths <35 weeks received 3 doses of 8 mg in the week before birth.

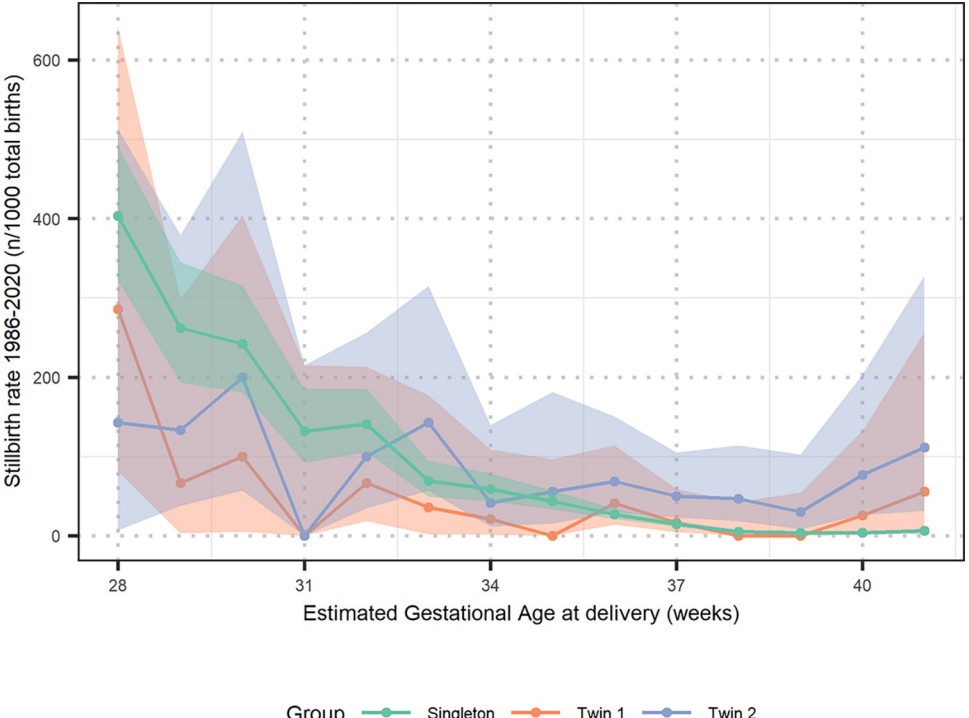

**Fig 2. Stillbirth rate for twin 1, twin 2 and singleton 1986–2020.** Shaded region is the 95% confidence interval for the estimated rate.

## Maternal death

There were 0.5% (3/597) maternal deaths recorded in mothers with a twin pregnancy with an EGA ≥28 weeks compared with 0.11% (65/59,005) in mothers with a singleton pregnancy with an EGA ≥28 weeks (Table 5). There was no improvement of maternal death over three decades. (S6 Table in the S1 File) The maternal death of the twin pregnancies were due to PPH (n = 1), pulmonary embolism (n = 1) and unknown reasons as it occurred during sleep (n = 1). The cause of maternal deaths of the singleton pregnancies were: PPH (n = 16), sepsis/ infection (n = 12), eclampsia (n = 11), malaria (n = 6), uterine rupture (n = 5), APH (n = 5) and unknown (n = 1).

## Discussion

This study cohort of twin pregnancies in a LMIC setting is unique, filling a data gap for marginalized migrants and refugees. Twin pregnancy in these populations presents a significant risk for mother and babies: increasing mortality fourfold for mothers and more than two-fold for babies. Risk of maternal morbidity for twin mothers is more than doubled for pre-eclampsia, preterm birth, and PPH compared to singleton pregnancies. These findings are consistent with the international literature [5]. In data from 23 LMICs, pre-eclampsia was diagnosed in 1.1% of twin pregnancies compared to 0.5% of the singletons, and preterm birth affected 35.2% of twin pregnancies compared to 9.6% of singletons [5]. In this study pre-eclampsia and preterm birth were diagnosed in 7.1% and 44% of twin pregnancies and 1.7 and 9.4% of singletons respectively i.e. more than a four-fold increased risk of complications in twin pregnancies. While the proportion of pre-eclampsia is higher it is not however as high as reported in other studies [34].

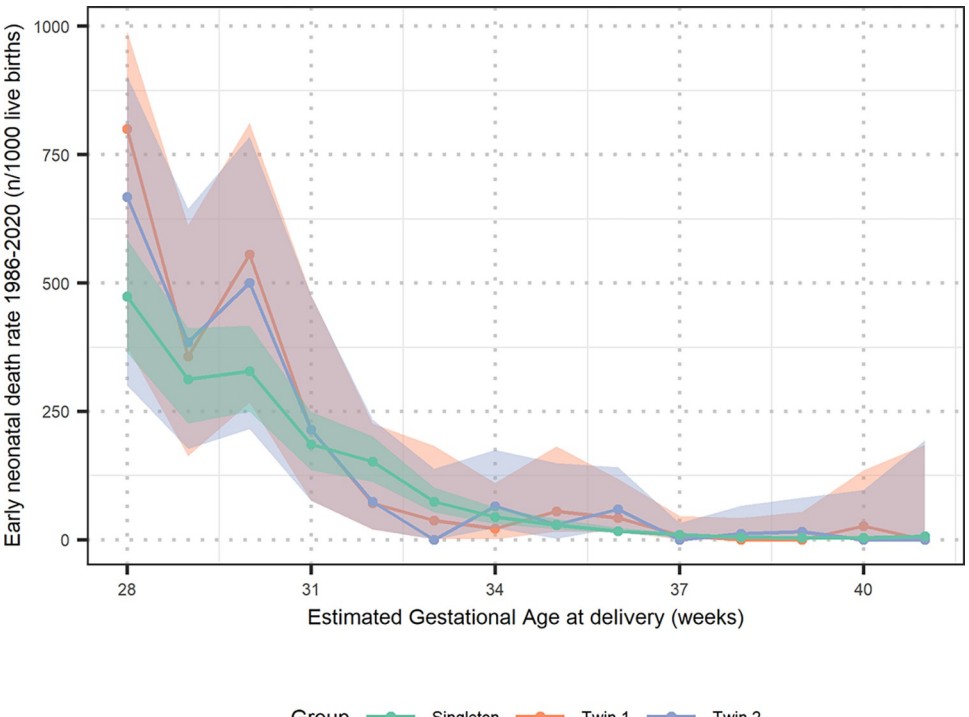

**Fig 3. Early Neonatal Death rate for twin 1, twin 2 and singleton 1986–2020.** Shaded region is the 95% confidence interval for the estimated rate.

In our cohort the natural twinning rate was around 5–10 per 1,000 births, lower than the worldwide rate of 12 per 1,000 births [1]. A low natural twinning rate <9 per 1,000 births was reported in East-Asia, including Thailand, while the stable rate across the study timeframes is reflective of young maternal age and the level of healthcare available to marginalized groups where there are no assisted fertility treatments [35].

In this cohort 75% of the twin pregnancies had a vaginal birth. This is equal to other low resource settings where around 57–90% of women with twin pregnancies routinely have vaginal birth [5, 36]. On the contrary, migrant women in high resource settings have higher caesarean rates compared to the non-migrant population, partly because of suboptimal healthcare through barriers as for example language [37, 38]. In this study setting of migrant women, however, the caesarean delivery rates in singletons (4.0%) and twins (25.0%) were lower than the overall caesarean delivery rate of Thailand, which was 32.7% for Thai nationals in 2017 [39]. One hospital in Thailand reported a caesarean delivery rate around 58% for Thai nationals with twin pregnancies [40]. Caesarean delivery for multiple and singleton pregnancy has

**Table 5. Proportion of perinatal mortality over time for twin 1, twin 2 and singletons.**

| | Stillbirth rate (stillbirth/total births) | | | Early neonatal mortality rate (early neonatal death/ live births) | | |
|---|---|---|---|---|---|---|
| | Twin 1 | Twin 2 | Singleton | Twin 1 | Twin 2 | Singletons |
| 1991–2000 | 7 (1/137) | 44 (6/137) | 15 (208/13,584) | 74 (10/136) | 84 (11/131) | 13 (178/13,376) |
| 2001–2010 | 34 (8/232) | 69 (16/232) | 11 (241/21,982) | 49 (11/223) | 65 (14/216) | 10 (218/21,741) |
| 2011–2020 | 23 (5/219) | 68 (15/219) | 9 (196/21,954) | 23 (5/214) | 5 (1/204) | 7 (159/21,758) |

Data are rate per 1,000 (n/N)

been found to be associated with an increased risk of PPH and longer duration of recovery [41, 42]. We did not explore the difference between the mode of birth and PPH because the information of PPH at the referral hospital was not available to SMRU. For future births the inherent risks of a scarred uterus are amplified in a low resource setting where civil disruption may be a significant barrier to obtaining safe care in the next pregnancy. The marginalized population reported here are receiving cross-border care in a setting with high and growing rates of caesarean delivery in Thailand. Meanwhile, Myanmar is in the midst of another coup which seriously and negatively impacts access to maternal health services including contraception and delivery services, with implications for future safety for a woman with a uterine scar [43].

This setting has a high prevalence of SGA by international standards even in singleton pregnancies, as reported previously. This is most likely related to nutrition, helminth infection and maternal stature (average maternal height is 151cm) [44]. A high proportion, 17–32%, of SGA are reported for singletons in multiple countries in South East Asia [45]. These high rates of SGA may influence the overall caesarean delivery rate by decreasing the risk of cephalopelvic disproportion, but could increase caesarean delivery for the indication of fetal distress.

Access to hospital care for the marginalized populations reported here is limited due to a neglected health care system in Myanmar, distance, legal documentation, language barriers and financial resources. Twin births were associated with a higher perinatal mortality for both neonates compared to singletons, as expected. This is in line with other studies in low and high resource settings which show an increased association with perinatal mortality for vaginal birth in twin pregnancies [4, 5, 9, 36, 46]. However, in this cohort there was no indication that the doubling of the caesarean delivery rate over time resulted in lower stillbirth rates–in fact, there was a slight increase of stillbirths for twins over this time period. Overall, the stillbirth rate in twins in this cohort is similar to other low resource settings, around 40–60 per 1,000 births for twin pregnancies [5].

Our cohort shows that most of the early neonatal deaths regardless of singleton or twin status, are related to prematurity. Worldwide prematurity is the leading cause for perinatal mortality [47]. Most of the twins with early neonatal death did not receive dexamethasone within 7 days before birth for maturation of the fetal lungs due to arriving in established labor. Dexamethasone given within 7 days before birth could reduce neonatal mortality [48, 49]. The support of oxygen, intravenous fluid, antibiotics and nasogastric feeding have been available since 2008 in the SCBU for the support of preterm infants [29]. This support most likely influenced the observed decrease in the neonatal mortality [30].

The percentage of maternal death did not improve over time and is comparable to other low resource settings [5]. Previous data in this population have shown that screening and treatment decreased maternal death from malaria; and training of the staff in EmOC as in the ALSO course, decreased the proportion of PPH in maternal deaths over time [27, 50].

The data presented here with a higher odds of stillbirth with the second compared to the first twin is consistent with the published literature [51]. Our data shows no reduction of stillbirth for twin 2 over time. In failed vaginal birth of the 2nd twin the referral distance by car to the nearest hospital that can provide caesarean delivery impacts on the survival of these neonates [25]. Improved capacity through training on manual maneuvers has the potential to improve the outcome for vaginal births of the second twin when caesarean delivery is not easily accessible [52–54].

Adequate guidelines with measurement of the time interval and training for twin pregnancy and deliveries are lacking in multiple low resource settings, which is probably an important factor for a higher fetal and maternal morbidity and mortality. Creating adequate context specific guidelines, with a focus on anaemia, prevention of (pre-)eclampsia and training for twin deliveries, especially for obstetric emergencies can improve the maternal and fetal outcome

[55]. Further research should focus on reducing maternal and neonatal mortality. Implementing training in ultrasound chorionicity, improving use of antenatal corticosteroids, improving access to skilled attendants at birth, consideration of earlier induction of monochorionic twins, and upskilling in manual maneuvers for vaginal delivery of the aftercoming twin are directly actionable to reduce twin stillbirths.

This study shows a possible higher stillbirth proportion when the discordance in the weight of the twins is more than 40%. Discordance in weight is a possible sign of uteroplacental dysfunction and increases the risk of perinatal mortality [56]. There is weak evidence that caesarean delivery protects against stillbirth of the second twin if the second twin is 40% larger than the first twin [56, 57]. The relative risk on neonatal mortality when the discordance is more than 40% rate, is 1.6 (1.1–2.2) for vaginal birth compared to caesarean delivery [56]. The decision to do a vaginal delivery should be made case by case and should be influenced by the ultrasound and skills of the staff in handling obstetric emergencies.

One systematic review, from high resource settings with advanced healthcare, suggested that induction of twins should be considered from 36–37 weeks to prevent stillbirth (depending on chorionicity) [46]. However, in this setting there is limited neonatal care, with no advanced respiratory support possible, limited access to hospital care and not always precise gestational age estimation or information about chorionicity due to late first ANC visit. Our data does not have the chronicity included, but other data from Thailand suggest that around 50% of the twin pregnancies are monochorionic [58], and as suggested from a 50 year old publication from Burma (Myanmar) [59]. Our data suggest that induction around 37–38 weeks should be considered with the caveat that induction in a low resource setting compared to a high resource setting increases the risk of adverse outcomes, as there may be inadequate fetal and maternal monitoring [60]. However if diagnosis of chorionicity of the twin pregnancy and fetal monitoring is possible, induction of labour around 36–37 weeks EGA of monochorionic twin pregnancies is advised to prevent stillbirth [46].

The main limitation of this study is the observational and retrospective nature of the data although it presents the entire population cohort with few exclusions. The loss to follow-up was expected in this mobile, conflict affected and marginalized population. For more than 30 years all pregnant women have been encouraged to birth with skilled birth attendants with a reversal from 90% born at home to 90% born with skilled attendants [50]. A greater emphasis is placed on this in high risk situations such as multiple pregnancy which is a potential explanation for the difference in loss to follow up between twin and singleton pregnancies. Nevertheless the findings remain consistent with international literature indicating data robustness in this limited resource setting. Missing date are high for some variables for singleton and twin pregnancies, for Apgar scores (38–42%), blood transfusion (10–55%) and for singletons antepartum and postpartum haemorrhage (47–52%) which may bias these estimates of the morbidities and early neonatal mortality. However, the available data with a raised risk of morbidity in expected domains: such as anaemia, hypertensive disorders of pregnancy, and preterm birth in twin compared to singleton pregnancies suggests the available data is reliable and consistent with previous population and multi-country reviews [6, 61, 62]. Due to the difficulty to follow up women, maternal death and early neonatal death may be underestimated. Another limitation is ultrasound chorionicity in multiple pregnancy which was not available.

The strength of this study is the large unique cohort in a low resource setting in a marginalized population which provides useful information for future balanced counseling for women with multiple pregnancy where access to caesarean delivery can be difficult. It reinforces practical measures e.g. active management of delivery of twin 2, that can be improved for service providers. It helps direct training efforts for healthworkers supporting women from Myanmar given the dual setback of Covid-19 and the coup.

## Conclusion

This study reports a low rate of twin pregnancy in refugee and migrants on the Thai-Myanmar border but with higher maternal mortality and morbidity, and perinatal mortality, compared to singleton pregnancies. The stillbirth for twin pregnancies did not improve with time compared to a reduction for singleton pregnancies. This requires efforts to build local capacity to improve early attendance at antenatal care with sonographers trained in accurate chorionicity, prevention and detection of anaemia, hypertensive disorders and preterm labor, and in labor: safe and practical protocols for the second twin.

## Supporting information

**S1 File. S1-S6 Table.**
(DOCX)

## Acknowledgments

We would like to thank all the staff and volunteers but especially local staff who worked for SMRU and provided the free care for the migrant and refugee population.

## Author Contributions

**Conceptualization:** Taco J. Prins, Aung Myat Min.

**Data curation:** Taco J. Prins, Aung Myat Min, Mary E. Gilder, Nay Win Tun, Janneke Schepens, Kathryn McGregor, Verena I. Carrara, Jacher Wiladphaingern, Mu Koh Paw, Eh Moo, Julie A. Simpson, Chaisiri Angkurawaranon, Marcus J. Rijken, Michele van Vugt, François Nosten, Rose McGready.

**Formal analysis:** Taco J. Prins, Julie A. Simpson.

**Investigation:** Taco J. Prins.

**Methodology:** Taco J. Prins.

**Supervision:** Julie A. Simpson, Chaisiri Angkurawaranon, Marcus J. Rijken, Michele van Vugt, Rose McGready.

**Writing – original draft:** Taco J. Prins.

**Writing – review & editing:** Aung Myat Min, Mary E. Gilder, Nay Win Tun, Janneke Schepens, Kathryn McGregor, Verena I. Carrara, Jacher Wiladphaingern, Mu Koh Paw, Eh Moo, Julie A. Simpson, Chaisiri Angkurawaranon, Marcus J. Rijken, Michele van Vugt, François Nosten, Rose McGready.

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
