## [Decision Letter · Decision Letter 0]

24 Oct 2023

PONE-D-23-21143Perinatal outcome of twin pregnancy and mode of birth in marginalized migrant and refugee populations on the Thai Myanmar border: a population cohort.PLOS ONE

Dear Dr. Prins,

Thank you for submitting your manuscript to PLOS ONE. After careful consideration, we feel that it has merit but does not fully meet PLOS ONE’s publication criteria as it currently stands. Therefore, we invite you to submit a revised version of the manuscript that addresses the points raised during the review process.

Please respond to all reviewers comments point by point

We look forward to receiving your revised manuscript.

Kind regards,

Ahmed Mohamed Maged, MD

Academic Editor

PLOS ONE

Journal Requirements:

"JAS is supported by an Australian National Health and Medical Research Council Leadership Investigator Grant (#1196068). This project is partially supported by Chiang Mai University to TJP and CA. The others authors received no funding for this study. The Shoklo Malaria Research Unit is supported in part by the Wellcome-Trust Major Overseas Programme in Southeast Asia (grant number 220211, https://doi.org/10.35802/220211; lead applicant Nicholas Day). For the purpose of Open Access, the author has applied a CC BY public copyright licence to any Author Accepted Manuscript version arising from this submission."

Reviewers' comments:

Reviewer's Responses to Questions

**Comments to the Author**

1. Is the manuscript technically sound, and do the data support the conclusions?

Reviewer #1: Yes

Reviewer #2: Yes

2. Has the statistical analysis been performed appropriately and rigorously? 

Reviewer #1: Yes

Reviewer #2: Yes

3. Have the authors made all data underlying the findings in their manuscript fully available?

Reviewer #1: Yes

Reviewer #2: Yes

4. Is the manuscript presented in an intelligible fashion and written in standard English?

Reviewer #1: Yes

Reviewer #2: Yes

5. Review Comments to the Author

Reviewer #1: GENERAL OVERVIEW, ORIGINALITY

This paper retrospectively addresses the outcome of twin versus singleton deliveries in low- and middle-income countries with still some barriers to accessing hospital care, specifically in refugee and migrant clinics on the Thai Myanmar border between 1986 and 2020.

Even though we know that twin pregnancies and deliveries carry higher risks for maternal and neonatal morbidity and mortality, the paper implies interesting aspects since the care is still concentrated on a few essentials without the chance of careful prevention of problems.

APPROACH /RESULTS

TITLE: The title might also imply the comparison, the authors performed, e.g.:

“Comparison of perinatal outcome and mode of birth between twin and singleton pregnancies in migrant and refugee populations on the Thai Myanmar border: a population cohort”.

ABSTRACT: Clear

INTROCUCTION: Please use the term “Caesarean delivery” instead of “section” throughout the paper; alternatively, they might say only cesareans.

METHODS.

It is a pity, that the authors excluded deliveries > 28weeks although it has been shown that in vertex-first twins between 26 and 32 weeks a vaginal delivery is possible in 213/248 cases (86%) and a planned vaginal delivery had no negative impact on the outcome as compared to a primary Cesarean (Sentilhes L et al. Neonatal outcome of very preterm twins: policy of planned vaginal or cesarean delivery. AJOG 2015;213(1):73 e1- e7.6). Similarly, in the JUMODA-Study with 232 cases with planned vaginal delivery and 192 planned cesareans there were no significant differences in survival without detected morbidity: 80.8% versus 80.2% (Korb et al. Survival without severe neonatal morbidity in very preterm twins according to planned mode of delivery. AJOG 2021;Suppl.1 , A.16, p.9). The authors should better explain why they excluded all deliveries < 28 gestational weeks, although this is an important outcome per se.

It was for me not completely clear whether the stillbirth rate was determined in the whole population or only in those with an expected gestational age of > 28 gestational weeks, please express this explicitly.

It remains completely unclear whether the attending obstetricians had sufficient training to perform vaginal deliveries in twins. Was there any attempt to differentiate between MC and DC twin pregnancy? Please specify who finally attended the twin deliveries and whether there protocols, e.g. for how long did they wait when the second twin was in breech or transverse presentation? Were they trained to perform internal version and/or breech extraction in the second twin, e.g. Arabin B, Kyvernitakis I. Vaginal delivery of the second nonvertex twin: avoiding a poor outcome when the presenting part is not engaged. Obstet Gynecol. 2011;118(4):950-4?

Although the authors report on ethical approval, this might not even have been required in accordance with the guidelines of the working group for the survey and utilization of secondary data (AGENS).

Last but not least: Were the data stored only as paper charts or also electronically? Unfortunately, this is not clearly stated.

RESULTS

In Figure 1, the authors demonstrate the “Flow Chart” of data and patients lost to follow up. Can the authors explain the high rate of > 15 000 pregnancies with lost to follow up or unknown outcome among singleton pregnancies? Can they explain that in both cohorts around 40% of Apgar values were missing?

Table 1 compares characteristics of twin and singleton cohorts, Table 3 the mode of delivery of twin 1, 2 and singletons. The authors should also demonstrate significant differences between the cohorts in a further column.

The authors should also explain whether life born neonates had access to a neonatal intensive care unit (NICU).

In addition, it would be interesting to know the suspected reasons for maternal death in both cohorts given the serious consequences for the families and societies. The authors should also try to evaluate and whether there was an improvement over time in both cohorts.

ADEQUACY DISCUSSION

PRINCIPAL FINDINGS: When the authors briefly summarize the results, they already admit that there was no improvement of perinatal mortality over time within the twin cohort. This is a serious outcome because even in this population the outcome of singletons improved.

There might be several reasons, which should be debated and proposed to local health care politicians such as a) early ultrasound for all pregnant women to diagnose not only a multiple pregnancy but also chorionicity. The authors should debate that given the low overall twin rate of this cohort the relative contribution of MCDA or even MCMA twins might be high, e.g. up to nearly 50% as compared to Western populations of where it is only one third.

The fact that stillbirths at the end of pregnancy mainly occur in these MC twin pregnancies is well known, but the conclusions not to induce labor at about 36-37 weeks might be wrong if diagnosis by ultrasound and FHR monitoring is available.

Another serious aspect seems to be the lack of teaching how to perform vaginal deliveries of twins, which can be taught. The poor skills are mainly reflected in the relative increase of cesareans in the second twin (combined twin deliveries), which is frequently iatrogenic reflecting poor skills of the physicians in charge. Combined deliveries also carry the highest risks of poor maternal and neonatal outcome. Rapid manual delivery of the second twin is mandatory in all cases where the second twin is not in vertex position and it would be interesting to know the time interval between the delivery of the first and second twin. Teaching –possibly with mannequins- and measuring the time interval is even possible in poor settings and seems mandatory.

MEANING OF THE FINDINGS/ CLINICAL IMPLICATIONS

The authors should also define the weight difference between the twins, e.g. discordance in cases where both birth weights are available. From the published data, weak evidence may support consideration of cesarean delivery in extremes of discordance, from a practical standpoint, this may apply when the second twin is approximately >40% larger than the presenting co-twin (Christopher D et al: An Evidence-Based Approach to Determining Route of Delivery for Twin Gestations. Rev Obstet Gynecol. 2011;4(3/4):109-116) but also when it is largely smaller (more frequent).

RESEARCH IMPLICATIONS:

Prospective designs if available with consequent teaching efforts in early ultrasound and practical skills in vaginal delivery including pain treatment for manual maneuvers.

STRENGTH AND WEAKNESSES:

Strengths: As the authors, describe. It is interesting to see these results in settings without ART but also without any defensive medicine, where basic needs are the main issue.

Weaknesses: The impact for improvement politics by better access to prenatal care, ultrasound and skilled operators during birth is hard to evaluate in a retrospective setting but mandatory for the future.

CONCLUSIONS

The intention of this paper is to praise to mirror the outcome of both, singleton and twin pregnancies in a region with low income and a high rate of refugees. Therefore, it should be published. Several weaknesses are discussed and these aspects should be considered in a professional way. The conclusion of the paper should be very clear and strong to improve the care for both, the outcomes of mothers and their offspring in singleton and even more in twin pregnancies.

Reviewer #2: I am not sure that I misunderstand for the number of cases in table 3, please check the number of twin pregnancy (n=595) but in the detail: birthed vaginally is 597. Line 231, 2946 grams should be 2,946 grams.

6. PLOS authors have the option to publish the peer review history of their article (what does this mean?). If published, this will include your full peer review and any attached files.

Reviewer #1: No

Reviewer #2: No

---

## [Author Response · Author response to Decision Letter 0]

6 Dec 2023

Reviewers' comments:

Reviewer's Responses to Questions

Comments to the Author

1. Is the manuscript technically sound, and do the data support the conclusions?

Reviewer #1: Yes

Reviewer #2: Yes

2. Has the statistical analysis been performed appropriately and rigorously? 

Reviewer #1: Yes

Reviewer #2: Yes

3. Have the authors made all data underlying the findings in their manuscript fully available?

Reviewer #1: Yes

Reviewer #2: Yes

4. Is the manuscript presented in an intelligible fashion and written in standard English?

Reviewer #1: Yes

Reviewer #2: Yes

5. Review Comments to the Author

Reviewer #1: GENERAL OVERVIEW, ORIGINALITY

This paper retrospectively addresses the outcome of twin versus singleton deliveries in low- and middle-income countries with still some barriers to accessing hospital care, specifically in refugee and migrant clinics on the Thai Myanmar border between 1986 and 2020.

Even though we know that twin pregnancies and deliveries carry higher risks for maternal and neonatal morbidity and mortality, the paper implies interesting aspects since the care is still concentrated on a few essentials without the chance of careful prevention of problems.

APPROACH /RESULTS

TITLE: The title might also imply the comparison, the authors performed, e.g.:

“Comparison of perinatal outcome and mode of birth between twin and singleton pregnancies in migrant and refugee populations on the Thai Myanmar border: a population cohort”.

ABSTRACT: Clear

Good advice: amended

INTROCUCTION: Please use the term “Caesarean delivery” instead of “section” throughout the paper; alternatively, they might say only cesareans.

METHODS.

It is a pity, that the authors excluded deliveries > 28weeks although it has been shown that in vertex-first twins between 26 and 32 weeks a vaginal delivery is possible in 213/248 cases (86%) and a planned vaginal delivery had no negative impact on the outcome as compared to a primary Cesarean (Sentilhes L et al. Neonatal outcome of very preterm twins: policy of planned vaginal or cesarean delivery. AJOG 2015;213(1):73 e1- e7.6). Similarly, in the JUMODA-Study with 232 cases with planned vaginal delivery and 192 planned cesareans there were no significant differences in survival without detected morbidity: 80.8% versus 80.2% (Korb et al. Survival without severe neonatal morbidity in very preterm twins according to planned mode of delivery. AJOG 2021;Suppl.1 , A.16, p.9). The authors should better explain why they excluded all deliveries < 28 gestational weeks, although this is an important outcome per se.

line 162-166

The reason extreme prematurity (EGA <28 weeks) was excluded was because in this, as in other limited resource settings, there is only active management from an EGA of 28 weeks. A previous publication from the same population reviewing 21 years of birth outcomes from 22 to ≤28 weeks identified extreme prematurity as representing less than 1% of birth outcomes, a 1 year survival of less than 1% and an over-representation of twins[30]. 

It was for me not completely clear whether the stillbirth rate was determined in the whole population or only in those with an expected gestational age of > 28 gestational weeks, please express this explicitly.

Line 192

stillbirths per 1000 total births from EGA≥28 weeks.

It remains completely unclear whether the attending obstetricians had sufficient training to perform vaginal deliveries in twins. 

Please see the answer (question after next) on who delivered the twin.

Was there any attempt to differentiate between MC and DC twin pregnancy?

Line 174-175

Ultrasound examination for chorionicity in multiple gestation is not available (added to the methods), but this a good practice point for improvement. This is now included in the methods and has been added to the discussion as a limitation (line 437-438). 

 Please specify who finally attended the twin deliveries and whether there protocols, e.g. for how long did they wait when the second twin was in breech or transverse presentation? Were they trained to perform internal version and/or breech extraction in the second twin, e.g. Arabin B, Kyvernitakis I. Vaginal delivery of the second nonvertex twin: avoiding a poor outcome when the presenting part is not engaged. Obstet Gynecol. 2011;118(4):950-4?

Line 133-139

The birth units are midwife led with twin births co-supervised by an obstetric doctor, trained in multiple deliveries. The doctor is not present for the full 24 hours but may stay to assist if twin delivery is expected. Communication through radio was replaced by phone contact in 1996. In general after birth of the first twin, the second twin would be brought into a longitudinal position through external and rarely internal version if necessary. After twenty minutes or in cases requiring expedited delivery e.g. fetal distress, artificial rupture of the membranes and/or oxytocin would be commenced. 

Although the authors report on ethical approval, this might not even have been required in accordance with the guidelines of the working group for the survey and utilization of secondary data (AGENS).

Noted 

Last but not least: Were the data stored only as paper charts or also electronically? Unfortunately, this is not clearly stated.

Line 97-100

Paper-based records prior to 2008 were extracted manually to an electronic database so that ANC and birth outcome data could be retrieved electronically. Information from records such as the paper-based partogram or the hospital referral forms was retrieved from archives stored at SMRU when required for clarification. 

RESULTS

In Figure 1, the authors demonstrate the “Flow Chart” of data and patients lost to follow up. Can the authors explain the high rate of > 15 000 pregnancies with lost to follow up or unknown outcome among singleton pregnancies? 

Line 160-161

(a high loss to follow-up remains a constant feature of this mobile, conflict affected and marginalized population) 

As stated in methods (line 121-122), now for more than 30 years, all pregnant women have been encouraged to birth with trained midwives. 

The discussion has been amended to clarify the situation as follow:

Line 423-428

The loss to follow-up was expected in this mobile, conflict affected and marginalized population. For more than 30 years all pregnant women have been encouraged to birth with skilled birth attendants with a reversal from 90% born at home to 90% born with skilled attendants[50]. A greater emphasis is placed on this in high risk situations such as multiple pregnancy which is a potential explanation for the difference in loss to follow up between twin and singleton pregnancies. 

Can they explain that in both cohorts around 40% of Apgar values were missing?

Line 154-156

Midwifes of the SMRU were trained in Apgar and routinely assessed newborns from 1998 and this data was not provided on discharge from births in Thai Hospitals. 

An additional table with furth analysis of the place of birth was added as supplementary table 3

 Twin pregnancy Singleton pregnancy

 Twin 1 Twin 2 

SMRU 28 per 1000 (9/327) 67 per 1000 (22/327) 9 per 1000 (304/34,340)

Home/Other 61 per 1000 (5/82) 98 per 1000 (8/82) 7 per 1000 (97/13,574)

Hospital 6 per 1000 (1/177) 45 per 1000 (8/178) 33 per 1000 (158/4,749)

Table 1 compares characteristics of twin and singleton cohorts, Table 3 the mode of delivery of twin 1, 2 and singletons. The authors should also demonstrate significant differences between the cohorts in a further column.

For Table 1 we present the distribution of the maternal characteristics at first antenatal visit for twin and singleton pregnancies and comment qualitatively on any differences in the distributions, there is no hypothesis to test here and therefore we do no present p-values. This follows the recommendations of the STROBE guidelines for reporting of observational studies (see item 14a; https://journals.plos.org/plosmedicine/article?id=10.1371/journal.pmed.0040297 ). 

For Table 3, we present the percentage (n/N) of delivery type for twin 1, twin 2, and singleton, and comment qualitatively on the differences, this is not a hypothesis of interest to test and therefore we do provide p-values.

The authors should also explain whether life born neonates had access to a neonatal intensive care unit (NICU).

Line 150-154

In this setting, only oxygen, intravenous fluid, antibiotics, nasogastric feeding and phototherapy have been available in the special care baby unit (SCBU) from 2008 for the support of unwell preterm infants and reducing neonatal mortality by half [29]. SCBU improved the level of care available prior to 2008: oxygen, intramuscular antibiotics and nasogastric feeding. A neonatal intensive care unit (NICU) was not available routinely for this population.

In addition, it would be interesting to know the suspected reasons for maternal death in both cohorts given the serious consequences for the families and societies. The authors should also try to evaluate and whether there was an improvement over time in both cohorts.

Line 312-316

There was no improvement of maternal death over three decades. (S6 Table ) The maternal death of the twin pregnancies were due to PPH (n=1), pulmonary embolism (n=1) and unknown reasons as it occurred during sleep (n=1). The cause of maternal deaths of the singleton pregnancies were: PPH (n=16), sepsis/infection (n=12), eclampsia (n=11), malaria (n=6), uterine rupture (n=5), APH (n=5) and unknown (n=1) .

 % Maternal Death (Maternal Death/total births)

 Twin Singleton

1991-2000 0 (0/137) 0.1 (16/13,584)

2001-2010 0 (0/232) 0.1 (21/21,982)

2011-2020 1 (3/219) 0.1 (20/21,954)

ADEQUACY DISCUSSION

PRINCIPAL FINDINGS: When the authors briefly summarize the results, they already admit that there was no improvement of perinatal mortality over time within the twin cohort. This is a serious outcome because even in this population the outcome of singletons improved.

There might be several reasons, which should be debated and proposed to local health care politicians such as a) early ultrasound for all pregnant women to diagnose not only a multiple pregnancy but also chorionicity. The authors should debate that given the low overall twin rate of this cohort the relative contribution of MCDA or even MCMA twins might be high, e.g. up to nearly 50% as compared to Western populations of where it is only one third.

The fact that stillbirths at the end of pregnancy mainly occur in these MC twin pregnancies is well known, but the conclusions not to induce labor at about 36-37 weeks might be wrong if diagnosis by ultrasound and FHR monitoring is available.

Line 413-416

Our data does not have the chronicity included, but other data from Thailand suggest that around 50% of the twin pregnancies are monochorionic [58], and as suggested from a 50 year old publication from Burma (Myanmar)[59].

And line 418-420

However if diagnosis of chorionicity of the twin pregnancy and fetal monitoring is possible, induction of labour around 36-37 weeks EGA of monochorionic twin pregnancies is advised to prevent stillbirth[46].

Another serious aspect seems to be the lack of teaching how to perform vaginal deliveries of twins, which can be taught. The poor skills are mainly reflected in the relative increase of cesareans in the second twin (combined twin deliveries), which is frequently iatrogenic reflecting poor skills of the physicians in charge. Combined deliveries also carry the highest risks of poor maternal and neonatal outcome. Rapid manual delivery of the second twin is mandatory in all cases where the second twin is not in vertex position and it would be interesting to know the time interval between the delivery of the first and second twin. Teaching –possibly with mannequins- and measuring the time interval is even possible in poor settings and seems mandatory.

Line 388-390

Improved capacity through training on manual maneuvers has the potential to improve the outcome for vaginal births of the second twin when caesarean delivery is not easily accessible[52–54].

Line 395-399

Further research should focus on reducing maternal and neonatal mortality. Implementing training in ultrasound chorionicity, improving access to skilled attendants at birth, consideration of earlier induction of monochorionic twins, and upskilling in manual maneuvers for vaginal delivery of the aftercoming twin are directly actionable to reduce twin stillbirths.

MEANING OF THE FINDINGS/ CLINICAL IMPLICATIONS

The authors should also define the weight difference between the twins, e.g. discordance in cases where both birth weights are available. From the published data, weak evidence may support consideration of cesarean delivery in extremes of discordance, from a practical standpoint, this may apply when the second twin is approximately >40% larger than the presenting co-twin (Christopher D et al: An Evidence-Based Approach to Determining Route of Delivery for Twin Gestations. Rev Obstet Gynecol. 2011;4(3/4):109-116) but also when it is largely smaller (more frequent).

Thank you for this suggestion to look at the data in a constructive way.

Supplementary table 2: Stillbirth proportion (% (n/n [95%CI] ) in twin pregnancies and discordance of weight of Twin 1 and Twin 2 

 Stillbirth twin 1 Stillbirth twin 2

Weight difference Twin 1 and twin 2 less than 40% 1.3 (7/520) [0.5-2.8] 3.9 (20/520) [2.2-5.5]

Weight Twin 1 > 40% larger than twin 2. 0 (0/34) [NA] 14.7 (5/34) [4.4-28.1]

Weight Twin 2 > 40% larger than twin 1. 11.1 (2/18) [1.4-34.7] 5.6 (1/18) [0.1-27.3]

Line 282-284

Any discordance greater than 40% between twins resulted in an excess of stillbirth of the smaller twin.

Line 400-407

This study shows a possible higher stillbirth proportion when the discordance in the weight of the twins is more than 40%. Discordance in weight is a possible sign of uteroplacental dysfunction and increases risk of perinatal mortality[56]. There is weak evidence that caesarean delivery protects against stillbirth of the second twin if the second twin is 40% larger than the first twin [56,57]. The relative risk on neonatal mortality when the discordance is more than 40% rate, is 1.6 (1.1-2.2) for vaginal birth compared to caesarean delivery [56]. The decision to do a vaginal delivery should be made case by case and should be influenced by the ultrasound and skills of the staff in handling obstetric emergencies.

RESEARCH IMPLICATIONS:

Prospective designs if available with consequent teaching efforts in early ultrasound and practical skills in vaginal delivery including pain treatment for manual maneuvers.

Line 395-399

Further research should focus on reducing maternal and neonatal mortality. Implementing training in ultrasound chorionicity, improving access to skilled attendants at birth, consideration of earlier induction of monochorionic twins, and upskilling in manual maneuvers for vaginal delivery of the aftercoming twin are directly actionable to reduce twin stillbirths.

STRENGTH AND WEAKNESSES:

Strengths: As the authors, describe. It is interesting to see these results in settings without ART but also without any defensive medicine, where basic needs are the main issue.

Weaknesses: The impact for improvement politics by better access to prenatal care, ultrasound and skilled operators during birth is hard to evaluate in a retrospective setting but mandatory for the future.

CONCLUSIONS

The intention of this paper is to praise to mirror the outcome of both, singleton and twin pregnancies in a region with low income and a high rate of refugees. Therefore, it should be published. Several weaknesses are discussed and these aspects should be considered in a professional way. The conclusion of the paper should be very clear and strong to improve the care for both, the outcomes of mothers and their offspring in singleton and even more in twin pregnancies.

Conclusion is amended line 448-454

Reviewer #2: I am not sure that I misunderstand for the number of cases in table 3, please check the number of twin pregnancy (n=595) but in the detail: birthed vaginally is 597. Line 231, 2946 grams should be 2,946 grams.

Table 3 Thank you and amended to n=597 line 257 and in line 262 changed in 2,946

6. PLOS authors have the option to publish the peer review history of their article (what does this mean?). If published, this will include your full peer review and any attached files.

Do you want your identity to be public for this peer review? For information about this choice, including consent withdrawal, please see our Privacy Policy.

Reviewer #1: No

Reviewer #2: No

---

## [Decision Letter · Decision Letter 1]

13 Mar 2024

Comparison of perinatal outcome and mode of birth of twin and singleton pregnancies in migrant and refugee populations on the Thai Myanmar border: a population cohort.

PONE-D-23-21143R1

Dear Dr. Prins,

We’re pleased to inform you that your manuscript has been judged scientifically suitable for publication and will be formally accepted for publication once it meets all outstanding technical requirements.

Kind regards,

Ahmed Mohamed Maged, MD

Academic Editor

PLOS ONE

Additional Editor Comments (optional):

Reviewers' comments:

Reviewer's Responses to Questions

**Comments to the Author**

1. If the authors have adequately addressed your comments raised in a previous round of review and you feel that this manuscript is now acceptable for publication, you may indicate that here to bypass the “Comments to the Author” section, enter your conflict of interest statement in the “Confidential to Editor” section, and submit your "Accept" recommendation.

Reviewer #1: All comments have been addressed

2. Is the manuscript technically sound, and do the data support the conclusions?

Reviewer #1: Yes

3. Has the statistical analysis been performed appropriately and rigorously? 

Reviewer #1: Yes

4. Have the authors made all data underlying the findings in their manuscript fully available?

Reviewer #1: Yes

5. Is the manuscript presented in an intelligible fashion and written in standard English?

Reviewer #1: Yes

6. Review Comments to the Author

Reviewer #1: The authors have now considered the aspects previously criticized and improved the manuscript.

To conclude, the manuscript should now be considered for publication.

7. PLOS authors have the option to publish the peer review history of their article (what does this mean?). If published, this will include your full peer review and any attached files.

Reviewer #1: No

---

## [Editor Report · Acceptance letter]

22 Mar 2024

PONE-D-23-21143R1 

PLOS ONE

Dear Dr. Prins, 

I'm pleased to inform you that your manuscript has been deemed suitable for publication in PLOS ONE. Congratulations! Your manuscript is now being handed over to our production team.

Kind regards, 

on behalf of

Professor Ahmed Mohamed Maged 

Academic Editor

PLOS ONE